# Explanation of Experimentally Observed Phenomena in Hot Tokamak Plasmas from the Nonequilibrium Thermodynamics Position

**DOI:** 10.3390/e22010053

**Published:** 2019-12-30

**Authors:** Ksenia A. Razumova, Valerii F. Andreev, Nadezhda V. Kasyanova, Sergey E. Lysenko

**Affiliations:** 1Kurchatov Complex of Thermonuclear Energy and Plasma Technologies, NRC ‘Kurchatov Institute’, 123182 Moscow, Russiavfandreev@gmail.com (V.F.A.); kasyanova@phystech.edu (N.V.K.); 2General Physics Chair, Moscow Institute of Physics and Technology, Dolgoprudny, 141701 Moscow, Russia

**Keywords:** energy transport, tokamak plasma confinement, plasma self-organization

## Abstract

In studying the hot plasma behavior in tokamak devices, the classical approach for collisional processes is traditionally used. This approach leaves unexplained a number of phenomena observed in experiments related to plasma energy confinement. Further, it is well known that tokamak plasma is always turbulent and self-organized. In the present paper, we show that the nonequilibrium thermodynamics approach allows us to explain many observed dependences and paradoxes; for example, puffing of impurities results in confinement improvement if zones of plasma cooling by impurities and additional plasma heating are not overlapped. The analysis of the experimental results shows the important role of radiation losses at the plasma edge in the processes determining its total energy confinement. It is shown that the generally accepted dependence of energy confinement on plasma density is not quite adequate because it is a consequence of dependence on radiation losses. The phenomenon of the appearance of internal transport barriers and magnetic islands can also be explained by plasma self-organization. The obtained results may be taken into account when calculating the operation of a future tokamak reactor.

## 1. Introduction

For more than 60 years, the behavior of hot plasma in tokamak devices (toroidal chamber with a strong longitudinal magnetic field and longitudinal current) has been studied. The aim of such research is to create a controlled thermonuclear fusion reactor that does not produce radioactive waste and does not pollute the atmosphere. The relevance of the problem has grown over the years.

Many years of experiments with tokamak devices show that plasma is always turbulent and cannot be described by classical equations for collisional processes, even if we increase the transport coefficients up to some anomalous value. Many interesting phenomena and regimes have been observed. The library of so-called “modes of operation” includes L-mode, H-mode, RI-mode, Super Shot, Advanced Tokamak, Internal Transport Barrier (ITB), and so on.

Many authors have tried to build some fitting models that can describe plasma behavior in some area of experimental conditions. In these models, the flux depends on the gradient of local plasma parameters. However, it appears that for each tokamak and each experimental condition, different models are needed.

In some models, a new term, the so-called “pinch flux”, appeared, which is the heat flux to the plasma center enhancing the classical value. This helped in some cases but did not solve all the problems.

It was shown (see, for example, [1]) that many nonlocal phenomena, for example, the energy confinement improvement associated with edge radiation cooling, cannot be described by means of the local flux-gradient expressions.

Powerful modern computers gave hope for calculating processes in plasma from the first-principles approach. The mighty gyrokinetic codes that had to calculate the behavior of all instabilities were created. However, plasma has many degrees of freedom and numerous instabilities and waves, which also interact nonlinearly with each other, and it is impossible to take into account all these badly known processes. Even the best modern computers do not allow us to perform the calculations to describe the magnetohydrodynamic (MHD) processes.

Scientists had to provide the information for the experimental reactor project. They collected much experimental information, but without the possibility of explaining processes by physics, they turned to building a scaling—empirical dependence of the energy confinement time $\tau_{E}=W_{d}/P_{in}$ on the discharge parameters. Here, *W_d_* is stored plasma energy and *P_in_* is injected heating power. The international thermonuclear reactor ITER was designed and built on the base of this scaling [2], but any scaling is only valid in the parameter range where it is found. In our case, the scaling requires extensive extrapolation—more than one order of value. This may be not so dangerous for independent external parameters, but the scaling includes an internal parameter, the plasma density, which depends on *τ_E_* itself. Besides that, we need scalings for transition from one mode of operation to another. Difficulties would strongly decrease if we could understand the physics of the processes that are determined by turbulence behavior.

We offer another approach to explain the abovementioned experimental data, which is based on the idea of plasma self-organization. It is well known that tokamak plasma is turbulent and self-organized [3,4,5,6,7]. It was shown experimentally that the normalized pressure profile $p_{N}(\rho )$ is independent of any parameters. Here, $\rho =r/{\left(IR/kB\right)}^{1/2}$, where *I* is the plasma current, *B* is the longitudinal magnetic field, *k* is the plasma cross section elongation, and *r* is the minor tokamak radius. In contrast with collisional systems, in turbulent plasmas, the particles and their energy cannot pass to the wall with different speeds. Plasma escapes by clusters, so there is no diffusion, only thermal conductivity. An energy balance equation has to hold constant the pressure profile. So, we need to find such an energy balance equation that can support the normalized pressure profile *p_N_*(*ρ*). In [8,9], Dyabilin and Razumova suggested using a Smoluchowski-like equation, which can maintain a given self-organized pressure profile.

In this work, the explanation of the observed experimental phenomena from the plasma self-organization position is offered.

## 2. Self-Organization

The self-consistent profile phenomena can be interpreted in the frame of a thermodynamic approach, when the self-consistent solutions correspond to the minimum of the free energy:(1)F=−θS+E
where *S* and *E* are entropy and energy, respectively, and *θ* is some effective temperature. Following the terminology from [5], we call parameter *θ* the “magnetic temperature”.

For the energy balance, we have [8]
(2)$$\frac{\partial p}{\partial t}=\nabla \cdot \left[\frac{\theta p}{\xi }\nabla \ln(p/p_{c})\right]+Q_{s}$$
where *p*(*r*) is the plasma pressure, *p_c_*(*r*) is the self-organized plasma pressure, *Q_s_* is a source of heating and radiation cooling, ξ in the Smoluchowski equation describes dissipation in the turbulent system, and the transport coefficient (thermal conductivity) κ=θ/ξ. The heat flow density is given by
(3)$$\Gamma =\kappa p\left(\left|\frac{\nabla p}{p}\right|-\left|\frac{\nabla p_{c}}{p_{c}}\right|\right).$$

In the general case, we need to add the neoclassical flux into Equations (2) and (3). For electrons, it is small compared with turbulent flux, but for ions, sometimes it can be significant.

In [8,9], Dyabilin and Razumova considered the magnetic confinement system in general and obtained pressure profiles that depended on some parameter *γ*. Comparison of these profiles with those measured experimentally in tokamak experiments shows with good accuracy that for tokamaks $\gamma \propto q_{L}$, *θ* depends on external discharge parameters and the plasma stored energy: $\theta \propto p_{0}\beta_{0}/q_{L}$, where *p*_0_ is the plasma pressure in the center, $\beta_{0}=8\pi p_{0}/B^{2}$, and *q_L_* is the safety factor at the edge.

The first term in brackets in Expression (3) corresponds to the outward gradient flow, and the second term characterizes the thermal pinch, which depends only on the shape of the self-organized profile. The external source of heat and loss terms *Qs*, distorting the pressure profile, leads to an increase in the level of turbulence and, in turn, increases the transport coefficients. Changes induced by turbulence take place on a short time scale. Experimentally, this is demonstrated, for example, by the speed of the heat wave that propagates along the minor radius after the electron cyclotron resonance (ECR) heating is switched on/off. The time of the heat pulse propagation is two orders shorter than the plasma energy confinement time [10]. The stronger the external influences on the plasma, leading to a distortion in the self-consistent pressure profile, the higher the free energy level (i.e., the turbulent activity), the greater the radial heat flux associated with the distortion of *p*(*r*), and the greater the transport coefficient κ=θ/ξ, as it depends on the level of turbulence.

Figure 1 shows that the normalized pressure profile *p_N_* within the limits of experimental errors is the same for tokamaks with different dimensions, heating methods, profiles, and intensity of heating. We see that the real *p_N_*(*ρ*) is always not far from the self-organized one *p_c_*(*ρ*). This means that the brackets in Equation (3) change slightly and the profile is not totally regulated by the gradient increase but by the change in ξ. The same follows from the calculations of specific experiments. However, even when there is no radial distorting heat flux (i.e., when the plasma free energy has a minimum), there still is a significant level of turbulence and, therefore, a significant value of energy transport.

The total heat flux Γ determined by the heating power deposited into the plasma can be represented as the sum of two fluxes:(4)$${\Gamma \;=\;\Gamma }_{0}{\;+\;\Gamma }_{1}$$
where Γ_0_ is the heat flux associated with the self-consistent pressure profile, where the turbulence level is minimal, and Γ_1_ is the part of flux Γ associated with the pressure profile distortion.

In order to accumulate the entire energy of additional heating in the plasma with the best confinement, we must realize a deposition profile that coincides with the profile of the self-consistent pressure. Then, Γ_1_ = 0, and it does not spoil the confinement. This option can be implemented more or less in ohmic mode, if the radiation loss does not significantly distort the profile of the deposited power. In such a case, the current density profile *j*(*r*) will be proportional to $T_{e}^{3/2}$, if the aspect ratio is not too low. Being self-organized, plasma has to form such *T_e_*(*r*) and *n_e_*(*r*) profiles, where *P_OH_*(*r*) coincides with *p_c_*(*r*). In this case,
(5)$$T_{e}^{3/2}(r)\propto n_{e}(r)\;T_{e}(r)\;\mathrm{or}\;T_{e}(r)\propto n_{e}{\left(r\right)}^{2}$$

This experimental result is presented in Figure 2. Being generated in the region of the local heating, Γ_1_ flows along the radius, distorting *p_c_*(*r*) and increasing transport. We can express the thermal conductivity coefficient as
(6)$$\kappa =\theta \;\left(\chi {\;}_{0}+\chi {\;}_{1}{(\Gamma }_{1})\right),$$
where $\chi =1/\xi =\chi_{0}+\chi_{1}$ and θ χ has a dimension of diffusion.

## 3. Most Unclear Experimental Fact: RI mode

Among the many unexplained phenomena is the dependence of the energy confinement time $\tau_{E}=W_{d}/P_{in}$ on the plasma density observed experimentally (Figure 3) [11]. At low densities, $\tau_{E}\propto n_{e}$, then it saturates. It is included in the ITER scaling. It seems to be very important, since the physical cause of this dependence remained unclear.

To understand this phenomenon, special experiments were performed in the T-10 tokamak (major radius of toroidal plasma *R* = 1.5 m, minor radius *a* = 0.3 m), where, in addition to ohmic heating (OH), plasma was heated by ECR power [12]. It has been shown that in conditions where the plasma density is maintained constant by a feedback system (Figure 4), the stored energy *W_d_* is independent of the plasma density (Figure 4a); *W_d_* may be increased by 40% due to the Ne injection with a constant density (within 10%). In Figure 4b, the stored energy *W_d_* is shown as a function of radiation losses *P_rad_* for the same shots. The form of this curve is typical for the $\tau_{E}(n_{e})$ dependence shown in Figure 3. After the initial increase of $\tau_{E}(P_{rad})$ in the left part of the curve (while the plasma density is the same), it becomes independent of *P_rad_*. Thus, we observed that the plasma stored energy is strongly dependent on radiation losses and nearly independent of the plasma density.

The effect of confinement increases with the increase of radiation losses after different gas injections into the plasma was discovered in the 1990s by Ongena et al. and Messiaen and Ongena [13,14]. This phenomenon has been named “RI-mode”.

In Figure 5, the dependence of *P_rad_* on the density for the OH case is presented. In the main part of the curve, *P_rad_* is proportional to *n*_e_. Why? The injection of the main working gas H_2_ or D_2_ leads to a strong charge-exchange flux to the wall and an increase of the light impurity influx into the plasma. For H_2_, it has to be less intensive than for D_2_. Hence, when injecting working gas, we also injected light impurities. The intensity of the impurity flux depends on the charge-exchange flux, the kind of hydrogen isotope, and the wall conditions.

We see that *τ_E_* is independent of plasma density. In contrast, however, the plasma density obviously depends on plasma energy confinement as energy and density in a turbulent plasma move to the wall with the same speed. How we can explain *τ*_E_ dependence on *P_rad_*?

If the radiation loss *P_rad_* reduces the flux Γ_1_ (Equation (3)), it should lead to a decrease in $\chi_{1}(\Gamma_{1})$ in the region where cooling takes place and outside it. Hence, the confinement improves because the plasma is self-organized, and an increase in the edge gradient ∇*p* means its growth over the whole plasma cross section. When the increase of the radiation losses leads to $P_{rad}/(2\pi R\;2\pi r)\approx \Gamma_{1}$, then $\chi_{1}(\Gamma_{1})$ will be small, and the improvement in confinement with increasing *P_rad_* saturates. A further increase in *P_rad_* will lead to a distortion of the pressure profile and the excitation of instabilities.

So, *χ*_0_ and *χ*_1_ in Equation (6) have somewhat different physical senses: *χ*_0_ is a value determined by the character of the turbulent flux, which plasma uses in its self-organization when the free energy is minimal. It cannot be decreased in any case and is independent of any parameters. *χ*_1_ depends on the distortion factor—the value and profile of the input power—which cannot be deposited into the self-consistent profile, but it is also independent of the other discharge parameters.

The coefficient κ=θ/ξ should be independent of local parameters *n_e_*(*r*), *T_e_*(*r*), and so on; otherwise, there can be no self-organization. All initial instabilities are important for turbulence development, but after the self-organization—mixing and averaging—the direct link between these instabilities and confinement disappears. It exists only for a time scale shorter than the turbulent relaxation time, which may be characterized by the heat/cold wave propagation time and is two orders of value less than *τ_E_*. The coefficient $\left(\chi_{0}+\chi_{1}(\Gamma_{1})\right)$ can be determined from experiments with various *P_in_*. Details of *χ*_0_ and *χ*_1_ determination are discussed in Section 5.

## 4. Dependence of Energy Confinement on Radiation Losses

It was shown in [14] that Ne, Ar, or Si injection or combinations yields the same improvement in energy confinement. It is important to check that this is valid also for He, which has *Z* = 2 and radiates at the very edge of the plasma. In T-10 experiments [12], it was shown that *W_d_* saturated at the same level for Ne and He injection for other similar conditions. Plasma in this tokamak was bounded with a circular metal limiter. It was possible, besides the usual working gas D_2_, to inject the noble gases Ne and He. Increasing the D_2_ injection, we also increased the fluxes of C and O from the wall. It was also possible to decrease the flux of these light impurities by lithiation of the wall and limiter surfaces, which strongly decreased the radiation losses.

Figure 6 shows the dependence of the energy confinement time *τ_E_* on the intensity of radiation losses for ohmic discharges with two different magnetic fields. The curves in Figure 6 may be divided into three parts. The first part for low *P_rad_* is characterized by a linear confinement increase with *P_rad_*; in the second part, we see independence of *τ_E_* from *P_rad_*, while too high *P_rad_* intensity leads to strong cooling of the plasma edge, pressure profile distortion, excitation of MHD instabilities, and *τ_E_* degradation. This effect is stronger for lower *q_L_* (shots with *B* = 1.9 T) because *p*_c_(*r*) distortion takes place near more important low q=m/n number rational surfaces.

According to the explanation given above, in the first part of the curve with linear increase of $\tau_{E}(P_{rad})$, the flux Γ_1_ distorting the plasma pressure profile is partly compensated by *P_rad_*. In the second part of the curve, the flux Γ_1_ is compensated totally in the radiation zone and outside it. In this case, Γ_1_ = 0 and *χ* is minimal; $\chi =\chi_{0}$. This is the best possible confinement (without the transport barrier, ITB). The difference in confinement for two series of shots is in accordance with $\theta (q_{L})$ dependence.

Once the increase in radiation losses leads to an improvement in confinement, a decrease in radiation losses should lead to degradation of confinement and loss of a significant fraction of the plasma. Using lithiation of the walls and limiter in the discharge chamber, it was possible to strongly decrease radiation losses. In the following experiments, the dependence of plasma confinement on radiation losses at the edge under strong additional ECR heating (by a factor of 8 greater than ohmic heating) and a wide change of *P_rad_* was studied. The scenario of the experiment was as follows: A series of shots with the same parameters was carried. At the ohmic stage of the discharge at time *t* = 500 ms, additional on-axis ECR heating with a power of 0.85 MW was used. At *t* = 600 ms, Ne was injected. The dependence of plasma confinement on the amount of injected impurities, that is, on the radiation value at the plasma edge, was studied.

In Figure 7, we show the results obtained with strong ECR heating (*P_EC_* = 0.85 MW). Without Ne (curve 1), the plasma stored energy *W_d_* was a little bit lower than for ohmic discharge. Applying Ne injection, as expected, we increased (restored) the stored energy. The stronger Ne injection led to a further improvement in the energy confinement (curves (2) and (3)).

An examination of these results with Equation (2) confirms the previous conclusion that the plasma energy confinement is independent of the plasma density (the initial densities for all shots were nearly the same) but depends on the radiation losses. An increase of Γ_1_ led to an increase in *χ*_1_, which, in turn, depended on Γ_1_. Ne injection reduced Γ_1_ in the outer part of the plasma and created an improved confinement zone. Although a small amount of Ne did not lead to a significant increase in the number of electrons in the plasma, the plasma density increased due to a decrease in *χ*_1_ at the edge.

As we need to use some gas injection for confinement increase, we have to know, should these ions accumulate in the hot core of plasma or it is possible to avoid this effect?

In ohmic discharges with low radiative losses in the plasma center, the profile of the input power has almost the same shape as the pressure profile, and the flux Γ_1_ is close to zero. Thus, for those conditions, the best energy confinement is obtained (in this case, *W_d_* does not increase with *P_rad_*). However, this also means that the pinch flux (second term in Equation (3)) is everywhere nearly comparable to the gradient flux, and impurities are able to penetrate into the hot plasma core. In such plasmas, the addition of a neoclassical flux may be important for ions. If their collision frequency is high, they may be not involved in turbulent movement and their confinement will be determined by usual collisional equations. In such a neoclassical case, ions will be distributed in plasma with a law: $n_{z}(r)\propto n_{p}^{z}(r)$, where *n_z_* and *n_p_* are densities of ions with charge *z* and protons correspondingly.

In the “clean” plasma at the beginning of the $\tau_{E}(P_{rad})$ dependence, *Z_eff_* = 1, but the energy confinement is poor. With increasing radiation losses at the edge, the pinch flow in Equation (3) will increase, the impurity ions will accumulate in the plasma core, and *Z_eff_* will increase. Its value depends on the intensity of the radiating gas injection. The highest *Z_eff_* will be reached when the plasma stored energy $W_{d}(P_{rad})$ reaches saturation due to the insertion into the plasma of more radiating ions; then, the energy confinement reaches its maximum. In this case, the radial distribution of the impurity ions and the working gas will be the same due to the specifics of turbulent transport (plasma participates in the motion “as a whole”). So, the effective plasma charge $Z_{eff}(r)=const$. This is supported by experimental results [15].

This may be a problem for reactors, since we will be forced to inject radiating impurities into the plasma volume, but we do not want to have strong bremsstrahlung radiation.

## 5. Calculations of *χ*_0_ and *χ*_1_

During the impurity gas puffing, the increase of radiation losses at the plasma edge results in a decrease of the radial heat flux Γ_1_ (up to Γ_1_ = 0); hence, the coefficient *χ* in the plasma edge reduces until κ reaches the minimum value $\kappa =\theta \;\chi_{0}$. As a result, the plasma stored energy increases and reaches the saturation level.

In order to find *χ*_0_ experimentally, we have to match the *χ*_0_ value in Equation (3) until the calculated stored energy $W=4\pi^{2}R\int_{0}^{a}\frac{3}{2}p(r)rdr$ coincides with the measured *W_d_*. As the additional heating power is measured with high precision and the area of the power input practically does not overlap with the area of the radiation losses, *χ*_0_ can also be calculated with sufficient precision. However, for ohmic discharges, where the power is deposited over the whole plasma cross section, the accuracy may be lower due to the less precise measurement of the radiation losses. Figure 8 shows measured value *χ*_0_ for T-10 experiments with Ne puffing, ECR heating, and ohmic heating. We see that within the experimental error, *χ*_0_ is independent of the heating power input.

Determination of $\left(\chi_{0}+\chi_{1}\right)$ is more complicated. The value of $\kappa =p_{0}^{2}IR(\chi_{0}+\chi_{1})/\left(B_{0}^{3}a^{2}\right)$ itself depends on $\Gamma =\Gamma_{1}+\Gamma_{0}$, as *χ*
_1_ depends on Γ_1_. For each case, we have to calculate Γ _0_, using the already known *χ*_0_, to find Γ_1_ and then match *χ*_1_(Γ_1_/Γ_0_) so that the given flux Γ corresponds to the measured value of *W_d_*. This allows us to obtain the value $k=\theta \left(\chi_{0}+\chi_{1}(\Gamma_{1}/\Gamma_{0})\right)$. If we do this for different heating fluxes, the dependence *χ* (Γ_1_/Γ_0_) may be obtained. The absolute values of *χ*_0_ and *χ*_1_ depend on the self-organized pressure profile *p_c_*, which we have to put into the Equations (2) and (3). As we do not have a reliable experimental profile, we used the theoretical profile from [8], shown in Figure 1.

## 6. Internal Transport Barriers (ITBs)

Local zones with enhanced confinement were discovered in tokamak plasmas at the beginning of the 1990s [16]. These zones are referred to as ITBs. Experimental investigation of the ITB has shown that in the ITB zone, the pressure gradients can largely exceed the gradient dictated by the profile consistency. So, existence of ITBs seems to be in contradiction with the self-consistent pressure profiles.

The following question naturally arises: what is the correspondence between profile consistency and ITBs? Are they two independent experimental facts, or are they in some way connected to each other?

From experimental results, it was concluded that a necessary condition for the appearance of the reduced transport zone is to have a low absolute value of dq/dr in the vicinity of the rational *q*-surface with low poloidal *m* and toroidal *n* numbers. The safety factor *q* characterizes a helicity of magnetic field lines, q=m/n. This result was found in experiments on many tokamaks: TCV [17], T-10 [18,19,20], and JET [21,22,23].

We know that rational surfaces have a specific feature. In principal, the magnetic surfaces fulfill all the space, but if we limit *m* by the upper number *m*_1_ (i.e., *m* < *m*_1_), then in the vicinity of the surface with low *m* and *n*, some area without magnetic surfaces exists, which is the so-called Gap. Inside this Gap, surfaces with *m* > *m*_1_ up to infinity exist, but there are no surfaces with *m* < *m*_1_. Simple analysis [20] shows that the radial width of the Gap is
(7)$$\delta_{\mathrm{gap}}\propto q/(m_{1}\;dq/dr).$$

Figure 9 shows the result of the rational surface density calculation for the MAST tokamak (low aspect ratio *R*/*a* = 1.5). We can see the electron temperature profile with ITB at *q* = 1 and rational surface density for two *m*_1_ values: *m*_1_ = 20 and *m*_1_ = 30. The lower *m*_1_ leads to a wider Gap. The turbulent heat flux experiences difficulties passing the Gap region. This means that the turbulent structure elements are linked with magnetic surfaces (may be with fluctuating current filaments). Cutting off the lower, the most powerful modes result in limiting the κ increase, more precisely, in *χ* increase, but *κ*, as we know, regulates the pressure profile near the self-consistent one for given Γ. If the external force distorts *p*(*ρ*), the brackets in Equation (3) must grow as a result of 1/*p*·*dp/dr* growth, and the free energy must grow too. In what form and in what structure will it appear in plasma?

To understand the conditions for ITB appearance, it is necessary to know which lower turbulence mode number *m*_1_ is needed for the radial transfer of heat flux. Using Relation (7) and comparing the calculated Gap width with the barrier width for the large number of published results for different tokamaks, it was possible to find a rough dependence linking the radial dependence of the heat flux density with *m*_1_. The result is shown in Figure 10 [20]. Despite the roughness of such an estimate (we do not know the radial correlation length of a given mode), the points obtained at a variety of tokamaks were well grouped on a steep dependence: *m*_1_ should decrease with increasing the flux density Γ, that is, $\Gamma \sim m_{1}^{-\alpha }$*,* where *3/2 < α < 5/2.* This means that when we increase Γ, the lower mode in the turbulent spectrum can get into the Gap near some rational surface, and if the derivative *dq*/*dr* is sufficiently low, it leads to the formation of a barrier.

Taking into account that for ITB formation we need to organize a wide Gap, we performed experiments, in which the ITB was generated in a given place with a given *q* [24]. The barrier near the surface with *q* = 3/2 was generated using off-axis ECR heating and a rapidly increasing plasma current. In Figure 11, the time evolution of the electron temperature *T_e_* for three plasma radii during ITB formation is given with three different time scales.

In Figure 12a, the scenario of the experiment is given. The ITB appeared after current ramp-up due to the reduction of *dq*/*dr* at *r* = 16 cm (Figure 12b). The temperature rose inside the zone with *r* = 16 cm. At *t* = 730 ms, the rise stopped and harmonic oscillations typical of the magnetic island with *m*/*n* = 3/2 appeared. The island grew, and at *t* = 763 ms, the first magnetic reconnection occurred with the release of energy outwards—this is the so-called “internal disruption”. This had to decrease the level of free energy. The process was repeated until the current noticeably penetrated from the periphery of the plasma inside its core, destroying the barrier.

So, we may conclude that the enhanced free energy (Equation (1)) at the top of the ITB leads to the formation of a new (fractal?) structure: the current island.

Our hypothesis is as follows: The heat flux coming from the inside increases ∇*p* in the Gap region. The heat flux supporting the self-consistent pressure profile should be based on pressure fluctuations extended along magnetic field lines. However, such fluctuations are associated with fluctuations in the bootstrap current, so we should expect the appearance of fluctuating current filaments. Overlapping in space, they can transfer the energy along the radius. The larger the flow, the larger the filaments (lower *m*) that should be generated; that is, the spectrum should expand toward low *m*. This is consistent with Figure 10. Deformation of the pressure profile in the Gap increases the free energy and decreases *m*, transforming the narrow filament into the current island.

We suppose that the radial flux is organized by small current filaments fluctuating in a radial direction. These filaments, accumulating free energy, appear due to deformation of the pressure profile in the Gap, forming current islands. Depending on the value of the stored free energy, they will have larger or smaller values of *m* and *n*, preserving the value q=m/n.

This has been well demonstrated in beautiful KSTAR experiments (Figure 13) [25]. In these experiments, the fine *T_e_* (and so, the current density) structure was registered from the image of the second EC harmonic emission. This gave the fine 2D temperature profile in some core plasma region with q≤1. Note that the current, flowing inside this region, forms a spiral with m/n=1/1 (kink mode), and the position of the EC resonance relative to the current center can be different. The geometry is complicated, and at various ECR heating positions relative to the current center, we may obtain *q* = 1 with *m*/*n* from 1/1 up to 3/3.

Similar results have been obtained in KSTAR experiments for small islands generated at the top of the external barrier (H-mode), the so-called edge localized modes (ELMs). Their amplitude and *m* number depend on the radial heat flux [26].

Energy overload of the islands cannot reduce *m* below the minimum value for a given surface (*m* = 1 in the KSTAR case shown in Figure 13). Further energy growth should lead to internal disruption, which is the mechanism that expels the energy through the barrier (see Figure 11 and Figure 12 as well as the KSTAR paper [25]). In this case, after the disruption, Γ(*r*) decreases in the region with *q* ≤ 1, and *m/n* changes from 1/1 to 8/8. Internal disruption is a new mechanism for maintaining the self-organized pressure profile. However, it is too rough and leads to the onset of large-scale MHD instabilities and global plasma disruption. Thus, the ITB, which initially leads to improvement in plasma energy confinement, is a precursor to its collapse.

## 7. Discussion

Plasma in the tokamak magnetic field is an open nonequilibrium system with many degrees of freedom. Such a system must be turbulent and self-organized, and it is not surprising that this is observed in experiments. The plasma system is well described from the position of nonequilibrium thermodynamics, giving the opportunity to move away from scalings and fitting models in fusion reactor design.

Following Dyabilin and Razumova [8,9], the energy balance equation was obtained, and the necessary transport coefficients were empirically found, which were determined only by external parameters and independent of the internal plasma parameters, such as temperature, density, effective charge, and so forth.

Dyabilin’s equation allows us to calculate any regime without an ITB for any tokamak, including the fusion reactor; experiments show that the physics processes are the same for all tokamaks, from the small RTP (*R* = 0.5 m, *a* = 0.16 m) to the large JET (*R* = 3 m, *a* = 0.9 m). We qualitatively explained the processes with internal barriers, but for quantitative calculations, the correct theory, such as that created by Dyabilin for a self-consistent pressure profile, is desirable. In [8,9], the functional was minimized only for diamagnetic currents because the experiment showed that the self-consistent pressure profile was independent of the longitudinal current density profile. To solve the problem with barriers, it is necessary to do similar work for longitudinal currents. However, the calculation of regimes with barriers is more difficult, since it requires knowledge of the longitudinal current density profile, which can be different and vary over time.

As shown above, in a fusion reactor, it is necessary to radiate a significant part of the generated power but in the plasma volume (radiation in the divertor volume cannot help). How can this be done without creating a significant concentration of radiating ions in the hot core, which can lead to unacceptable intensity of bremsstrahlung? We have to analyze this problem.

## 8. Conclusions

In this paper, on the basis of the thermodynamic approach, the phenomena of improved confinement at impurity influx and saturation of confinement dependence on the density in tokamaks were explained. A hypothesis was proposed to explain the formation of internal transport barriers and magnetic islands.

## Figures and Tables

**Figure 1 entropy-22-00053-f001:**
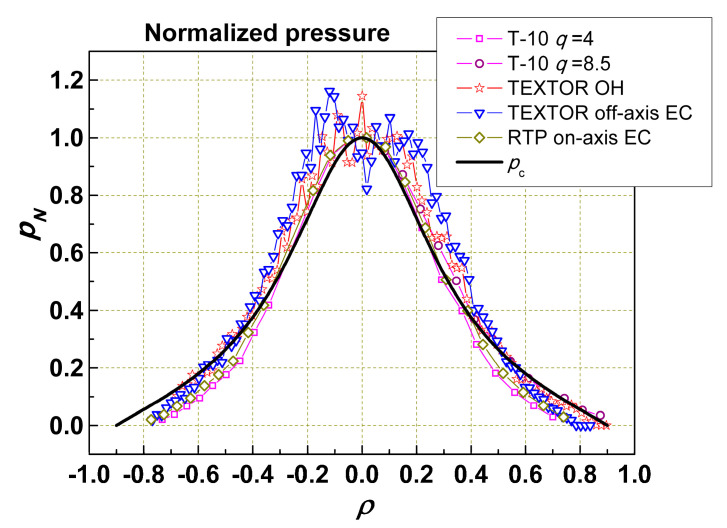
Normalized pressure profiles in different tokamaks [7] and self-consistent profile *p_c_* [8] calculated from thermodynamic approach vs normalized radius *ρ* = *r*/(*IR*/*B*)^1/2^.

**Figure 2 entropy-22-00053-f002:**
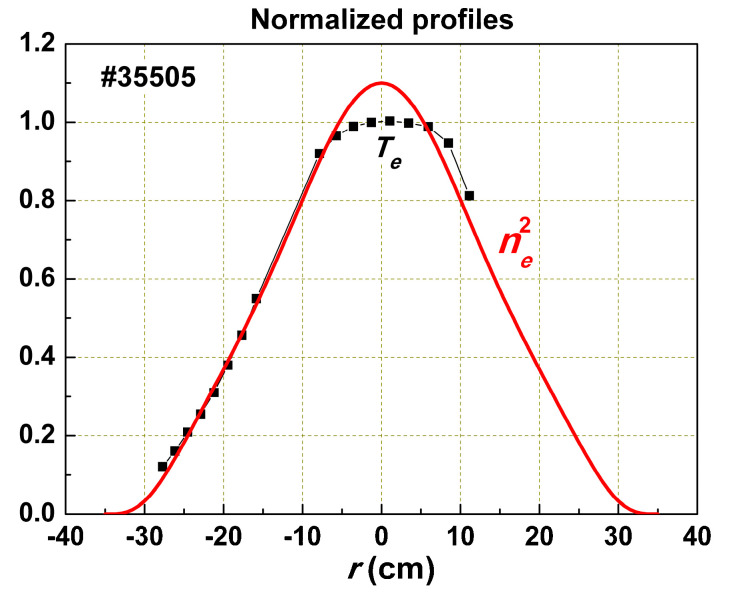
Normalized *T_e_*(*r*) and *n_e_*(*r*)^2^ profiles in T-10. In the central region, the *T_e_* profile is flattened due to sawtooth oscillations. The Abel inversion smooths this effect for *n_e_*.

**Figure 3 entropy-22-00053-f003:**
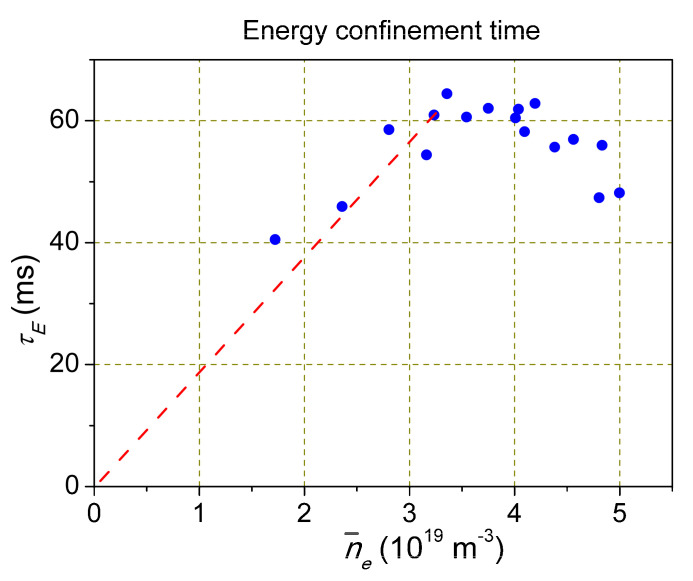
Typical dependence of the energy confinement time on plasma density [11]. Permission IoP Publishing.

**Figure 4 entropy-22-00053-f004:**
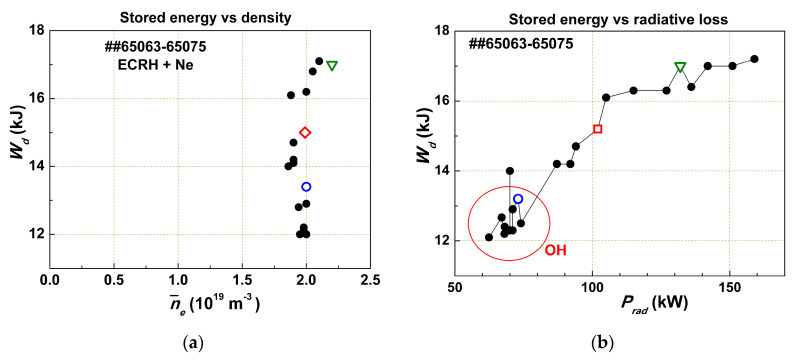
Results of electron cyclotron resonance heating (ECRH) and ohmic heating (OH) experiments with different amounts of injected neon: (**a**) steady-state values of the stored energy *W_d_* vs average plasma density; (**b**) *W_d_* vs radiative loss power *P**_rad_*. Some points are marked with symbols in order to track them in both figures [12] Reproduced with permission Pleiades Publishing.

**Figure 5 entropy-22-00053-f005:**
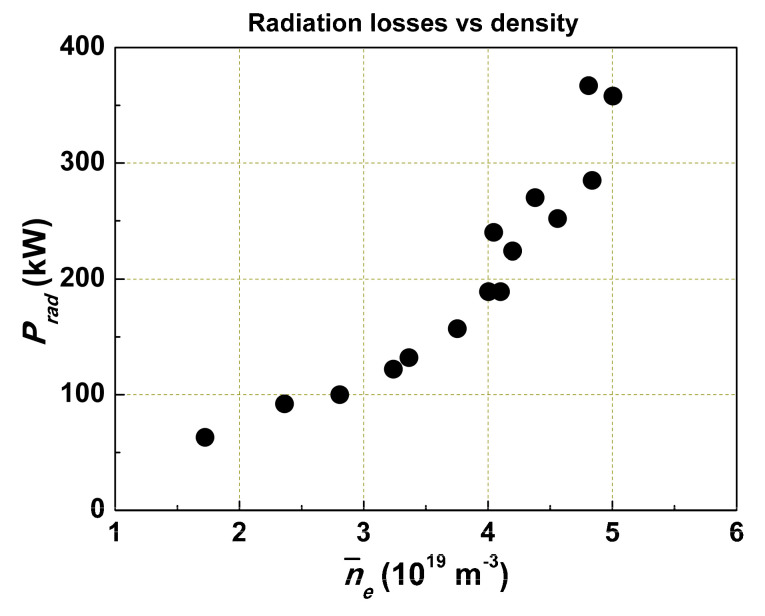
Dependence of radiative loss power on the density without Ne injection.

**Figure 6 entropy-22-00053-f006:**
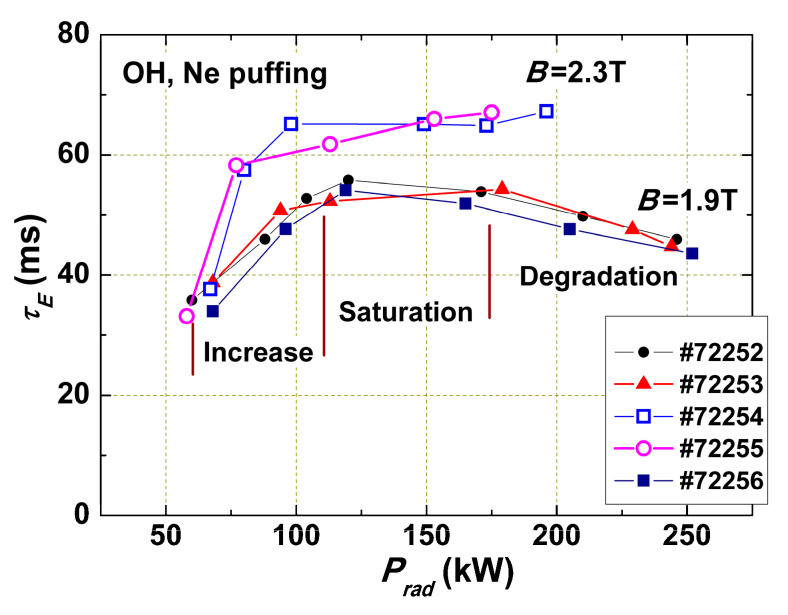
Dependence of the energy confinement time *τ_E_* on the total radiation loss power in ohmic shots with Ne injection: T-10, *I* = 230 kA, and *B* = 2.3 and 1.9 T. The lower series of shots was divided into parts with confinement increase, saturation, and degradation.

**Figure 7 entropy-22-00053-f007:**
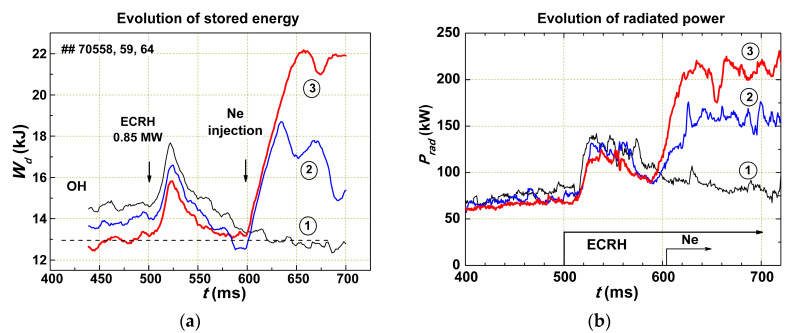
Time variation of the plasma stored energy *W_d_* (**a**) and radiation losses *P_rad_* (**b**) in experiments with ECR heating (*P_EC_* = 0.85 MW) and lithiation of the chamber walls. Curve (1) is for a discharge without Ne injection. Despite the powerful heating, the stored energy is lower than for the ohmic discharge. Curves (2) and (3) show the improvement of the energy confinement for shots with stronger Ne injection.

**Figure 8 entropy-22-00053-f008:**
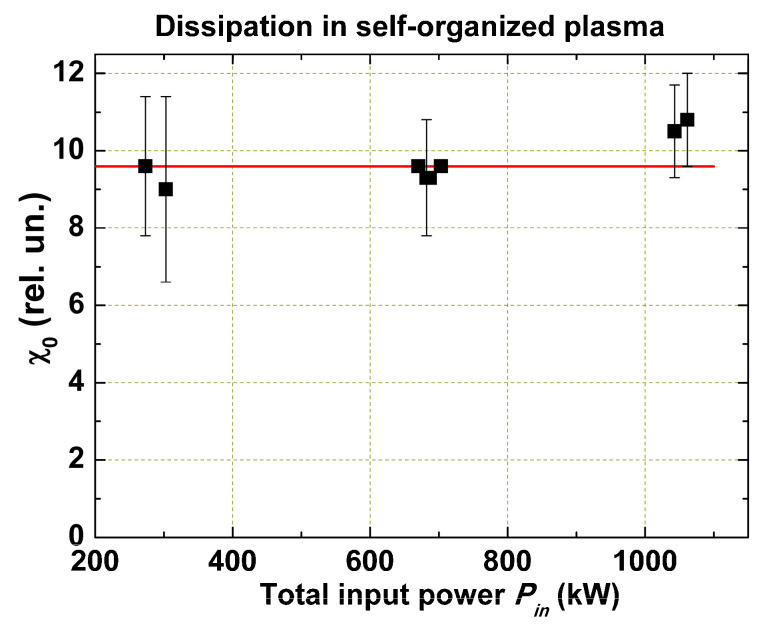
Measured value *χ*_0_ for experiments with Ne puffing, ECR heating, and ohmic heating.

**Figure 9 entropy-22-00053-f009:**
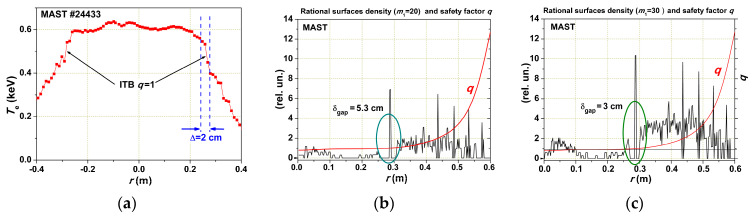
Experiment on the MAST tokamak: (**a**) the electron temperature profile with an internal transport barrier (ITB); (**b**,**c**) are results of rational surface density calculations for two different *m*_1_ values.

**Figure 10 entropy-22-00053-f010:**
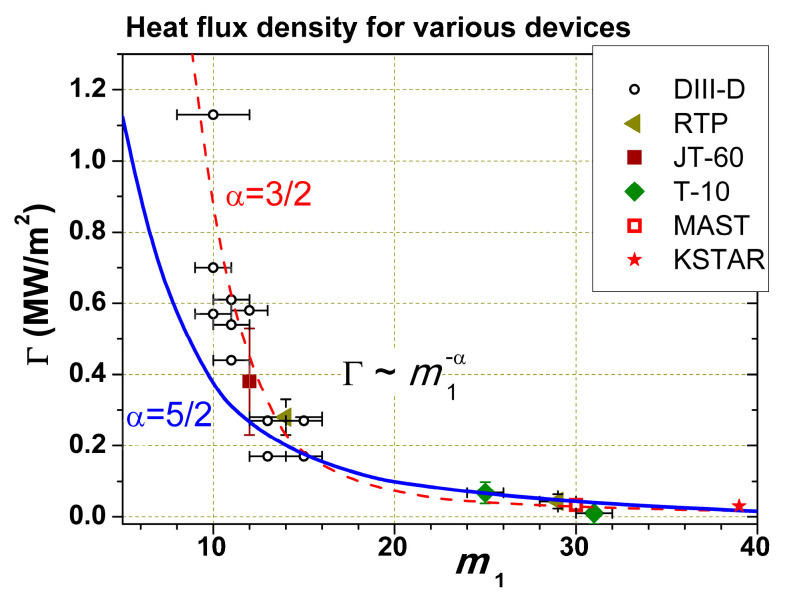
The relationship between the heat flux density Γ and the *m*_1_ number obtained from a comparison of calculated Gap and ITB widths.

**Figure 11 entropy-22-00053-f011:**
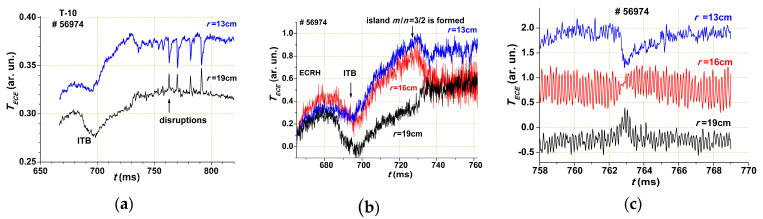
T-10 experiment with ITB formation. The time evolution of the electron temperature for three radii and three different time scales: (**a**) general evolution; (**b**) formation of ITB at *t* = 690 ms and magnetic island at *t* = 730 ms; and (**c**) internal disruptions after *t* = 760 ms.

**Figure 12 entropy-22-00053-f012:**
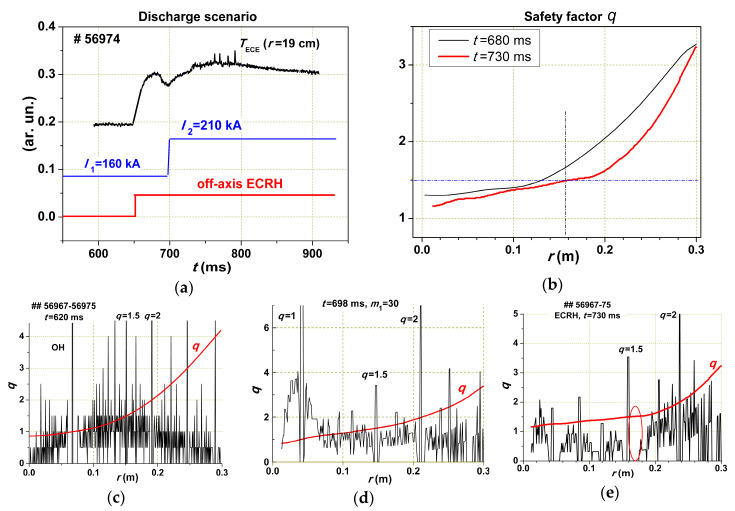
Experiment with ITB formation. (**a**) The time scenarios, from up to down: *T_e_* at *r* = 19 cm, the current rump-up, and ECR heating; (**b**) calculated safety factor profile before and after the current rump-up. Crossing of dash-dotted lines shows position of Gap at *q* = 3/2. (**c**–**e**) Densities of rational surfaces calculated for *m*_1_ = 30 and *q* profiles for different time moments: in OH phase, before current ramp-up, and after current ramp-up with ECR heating. In the latter case, a large gap marked by ellipse is seen.

**Figure 13 entropy-22-00053-f013:**
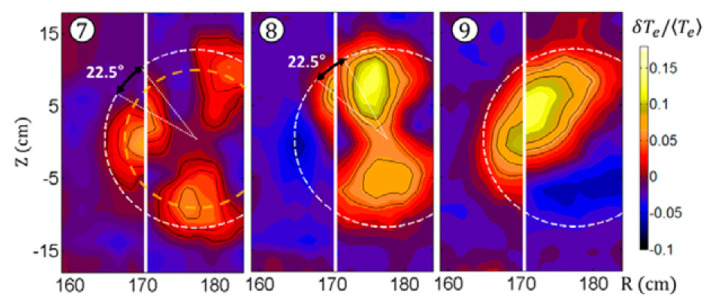
KSTAR experiment [25]. Islands with *m*/*n* = 3/3, 2/2, and 1/1 may exist interior to the region *q* = 1. Reproduced with permission IAEA.

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
