# Peer review of "Explanation of Experimentally Observed Phenomena in Hot Tokamak Plasmas from the Nonequilibrium Thermodynamics Position"

_entropy, 2019, doi:10.3390/e22010053_

Round 1

Reviewer 1 Report

This paper presents a new way to explain many observed trends and paradoxes experimentally observed in magnetically confined plasmas, based on a non-equilibrium thermodynamics approach. The important role of radiation losses is highligted while the generally accepted dependence of energy confinement on plasma density is shown to be inadequate. Other observed processes such as internal transport barriers and magnetic islands are also explained. The manuscript is clear and well written. The method used, as well as the results obtained are clearly presented. My opinion is that this paper can be expected to be of great interest for the plasma physics community and can be accepted in its present form.

Author Response

Dear Referee!

Many thanks for support of our views on transport in tokamaks.

We try to improve our English. We will be happy if you show us several   errors. Also we found some typos and made a small amendment in Discussion.

Our corrections are marked by yellow. Unclear issues for Editor are marked by blue.

Reviewer 2 Report

Dear authors,

I do recommend the manuscript to be accepted with minor revisions of the text. The presented field is of vast importance, the stability of plasma and bounded layer regions are crucial for the work of a tokamak type reactors. This is a step forward to relating the stability, as well as barrier forming question towards an consistent theory. The K.S. Dyabilin work is of importance as an example that the similar theory for the tokamak plasma without ITB could be related to the external parameters, and it is expected that the presented work could led to similar connection with theory. There is a tremendous both experimental as well as theoretical work behind the present state of modeling, and it is expected to have a step forward in near future. It was interesting fact for me that neon impurities affect in such significant way to the energy transfer since neon has a cross section for the ionization almost two times smaller than hydrogen, and related to the higher electron energies. The self-organized plasma of tokamak regions may be in relation with similar self organized phenomena in dusty or other types of strongly coupled systems.

Technical details follows

At line 56, reference [2] is in superscript, line 59 strange symbol before E, repeated at the line 262, please do check the entire text for the formulas nested into the text.This could be related to my lack of proprietary Windows fonts.

Figure 11 - please do consider resizing sub figures to be the same size, and if possible also consider enlarging most of the figures if possible in order to be more readable.

References are stated twice, some of them as supersrcript in the bibliography section.

Please re-check again the format of the paper, equations nested into the text, and figure sizes, the thematic and presented material is good, but the presentation is not.

Since it is common in Russian to use the phrase "very many" (очень много) and it is not quite common phrase in English, maybe use some of the synonym words such as "a lot of", "very much" or similar, but this is also a matter of personal decision. Previous remarks are crucial for the paper presentation quality.

Sincerely

Author Response

Dear Referee!

We took into account your comments and try to improve our English. Also we found several typos and amended several phrases in Discussion. Our corrections in the manuscript are marked by yellow. Unclear issues for Technical Editor are marked by blue.

 Below your comments start with R:, and our responses start with A:

R: At line 56, reference [2] is in superscript, line 59 strange symbol before E, repeated at the line 262

A: Corrected

R: Figure 11 - please do consider resizing sub figures to be the same size, and if possible also consider enlarging most of the figures if possible in order to be more readable.

A: Most of figures redrawn, sizes of numbers and letters are increased, also we used Bold letters. We will discuss resizing of subfigures with Technical Editor.  

R: References are stated twice, some of them as supersrcript in the bibliography section.

A: We did it intentionally, because we were unsure, how to treat Endnotes. Now it is corrected.

R:Please re-check again the format of the paper, equations nested into the text, and figure sizes, the thematic and presented material is good, but the presentation is not.

A: Formattig wil be discussed with Technical Editor after acception of manuscript

R: Since it is common in Russian to use the phrase "very many" (очень много) and it is not quite common phrase in English, maybe use some of the synonym words such as "a lot of", "very much" or similar, 

A: We decide to omit the word 'very' , so the ultimate text is  ' many degrees of freedom'.

Many thanks for support of our views on transport in tokamaks, and for careful reading.